# Macrophage-Derived Exosomal MALAT1 Induced by Hyperglycemia Regulates Vascular Calcification Through miR-143-3p/MGP Axis in Cultured Vascular Smooth Muscle Cells and Diabetic Rat Carotid Artery

**DOI:** 10.3390/cells14241995

**Published:** 2025-12-15

**Authors:** Kou-Gi Shyu, Bao-Wei Wang, Wei-Jen Fang, Chun-Ming Pan

**Affiliations:** 1Division of Cardiology, Shin Kong Wu Ho-Su Memorial Hospital, 95 Wen-Chang Road, Taipei 111, Taiwan; 2Department of Research, Shin Kong Wu Ho-Su Memorial Hospital, Taipei 111, Taiwan

**Keywords:** MALAT1, exosome, vascular calcification, macrophage, hyperglycemia

## Abstract

Metastasis-associated lung adenocarcinoma transcript 1(MALAT1) is associated with vascular calcification and diabetes-related complications. However, the effect of exosomal MALAT1 derived from macrophages induced by hyperglycemia on vascular calcification (VC) remains unclear. In this study, we investigated the effect of VC and its regulatory mechanisms in cultured vascular smooth muscle cells (VSMCs) and diabetic rats by exosomal MALAT1 derived from macrophages treated with high levels of glucose. Macrophages and VSMCs were cultured in 25 mM glucose. Macrophages exposed to high glucose exhibited increased expression of exosomal MALAT1. When transferred to VSMCs, exosomal MALAT1 significantly suppressed the expression of miR-143-3p while upregulating Matrix Gla protein (MGP, an inhibitor of VC) mRNA and protein levels. Interventions using MALAT1 siRNA or miR-143-3p mimics effectively reversed this effect. Both MALAT1 siRNA and overexpression of miR-143-3p significantly increased the calcium content in cultured VSMCs and in the carotid artery of diabetic rats following balloon injury. Balloon injury to the carotid artery in diabetic rats treated with macrophage-derived exosomes significantly increased the expression of MALAT1 and MGP while reducing the expression of miR-143-3p in the carotid artery. These findings demonstrate that macrophage-derived exosomal MALAT1 modulates VC via the MALAT1/miR-143-3p/MGP axis under hyperglycemic conditions. The results suggest that targeting exosomal MALAT1 may offer a novel and effective therapeutic approach for mitigating VC in metabolic disorders such as diabetes.

## 1. Introduction

Vascular calcification (VC) is characterized by the abnormal accumulation of calcium in the arterial wall, leading to arterial stiffness and impaired blood flow. VC plays a critical role in the progression of cardiovascular disease and increases the risk of cardiovascular morbidity and mortality [1]. Although the precise mechanisms of VC remain unclear, its incidence is rising annually, posing a significant threat to human health worldwide. To date, no effective prevention or treatment methods for VC exist, emphasizing the need for further research.

Hyperglycemia is a key factor in diabetes-associated vascular complications. It promotes VC and elevates the risk of arterial stiffness and cardiovascular events, such as myocardial infarction and stroke, particularly in diabetic patients [2,3]. The global prevalence and incidence of diabetes mellitus are rapidly increasing, and this condition is linked to higher morbidity and mortality from cardiovascular disease, posing a significant public health threat and contributing to a global pandemic [4,5]. Inflammation plays a central role in the VC process, with macrophages being the primary inflammatory cells involved. Hyperglycemia creates a pro-inflammatory environment that attracts macrophages to vascular sites. These macrophages contribute to the progression of VC [6]. The key mechanisms by which macrophages promote VC include increased oxidative stress, disrupted calcium–phosphate homeostasis, and inflammation [7].

Exosomes are extracellular vesicles of endosomal origin that play a critical role in intercellular communication [8]. Owing to their specific cargo content, exosomes hold promise as diagnostic markers for VC. Furthermore, they may serve as effective vehicles for therapeutic interventions, such as cargo modification to inhibit calcification. Prior research has demonstrated that exosomes play a pivotal role in VC [9,10]. Thus, investigating the relationship between exosomes and VC could deepen our understanding of potential therapeutic strategies for VC.

Exosomal long non-coding RNAs (lncRNAs) hold promise as biomarkers for detecting and monitoring VC, as well as therapeutic targets for interventions aimed at mitigating VC progression [11,12]. Metastasis-associated lung adenocarcinoma transcript 1 (MALAT1), a highly conserved lncRNA, may contribute to VC development by mediating the differentiation of vascular smooth muscle cells (VSMCs) into osteogenic-like cells capable of producing calcifying extracellular matrix components [13,14].

MicroRNA-143-3p (miR-143-3p) has been shown to be a target of MALAT1 [15,16]. Previous studies have reported that miR-143-3p has an important role in the regulation of cardiovascular calcification [17,18]. Matrix Gla protein (MGP) is an inhibitor of VC [19]. MGP has been shown to be the target of miR-143 [18].However, the MALAT1–miR143-3p-MGP axis in VC has not been reported. Furthermore, the effect of exosomal MALAT1 derived from macrophages treated with high levels of glucose on VC in VSMCs and balloon injury of the carotid artery in diabetic rats is not known. In this study, we investigated the effect of VC and its regulatory mechanisms in cultured VSMCs and diabetic rats by exosomal MALAT1 from macrophages treated with high levels of glucose.

## 2. Materials and Methods

### 2.1. Cell Culture

Macrophages (RAW264.7) were purchased from American Type Culture Collection (ATCC) and cultured in Dulbecco’s modified Eagle’s medium (DMEM, Thermo Fisher Scientific, Waltham, MA, USA) supplemented with 10% fetal bovine serum (FBS, Gibco, Thermo Fisher Scientific, Waltham, MA, USA) at 37 °C with 5% CO_2_. Macrophages were grown to 80–90% confluence in 10 cm^2^ culture dishes and were sub-cultured in the ratio of 1:3. For exosomes collection, 10 mL of cell-free supernatant from macrophages culture was collected after glucose 25 mM treatment.

Vascular smooth muscle cells (VSMCs) were obtained from PromoCell GmbH (Order No. c-12,511; Heidelberg, Germany). The cells were cultured in a smooth muscle cell growth medium (PromoCell GmbH (Order No. c-22,062), Heidelberg, Germany) supplemented with 10% fetal bovine serum, 100 U/mL penicillin, and 100 μg/mL streptomycin at 37°C in a humidified atmosphere with 5% CO_2_. VSMCs were grown to 80–90% confluence in 100 mm culture dishes.

### 2.2. Hyperglycemic Stress Environment

Cells were seeded in 100 mm dishes or on coverslips in cell growth medium with different glucose concentrations from 12.5 mM to 100 mM. The control glucose concentration is 5.5 mM.

### 2.3. Extraction of Exosomes from Cell Media

Exosome isolation from cell culture media was performed by Total ExosomeIsolation Reagent (Invitrogen, Thermo Fisher Scientific, Waltham, MA, USA) according to the manufacturer’s instruction. Briefly, cell-free supernatant was centrifuged at 2000× *g* for 30 min to remove cell debris. The supernatant containing the cell-free cell media was transferred to a fresh container and held on ice until use. Each sample was combined with 1/2 volume of Total Exosome Isolation Reagent and mixed well by vortexing or pipetting up and down until a homogenous solution was formed. The samples were incubated at 4°C overnight and then centrifuged at 4°C at 10,000× *g* for 1 h. The supernatant was aspirated and discarded, and the exosome pellet was re-suspended in 1X PBS buffer and then stored at 4°C for short term (1–7 days) or −20°C for long term. The exosome was quantitated by ExoQuantTM quantification assay kit according to the manufacture’s instruction (BioVision, Milpitas, CA, USA).

### 2.4. RNA Quality Assessment of Exosomes

We used Agilent 2100 Bioanalyzer to assess the quality of exosomes RNA with RNA Pico Chip (Agilent Technologies, Waldbronn, Germany) according to the protocol of the manufacturer. An electropherogram and a virtual gel image were generated for visualization and better interpretation. All chips were performed in duplicates.

### 2.5. Reverse Transcription and Real-Time Quantitative Polymerase Chain Reaction

We used High-Capacity cDNA Reverse Transcription Kits (Applied Biosystems, Thermo Fisher Scientific, Waltham, MA, USA) to quantify MALAT1-exosome RNA transcripts. Ten percent of each cDNA reaction was quantitated using standard SYBR chemistry and cycler conditions for quantitative polymerase chain reaction (qPCR).The Fast SYBR^®^ Green Master Mix (Applied Biosystems, Thermo Fisher Scientific) was applied for the further assays. The PCRs were performed on an ABI StepOnePlus cycler (Applied Biosystems, Thermo Fisher Scientific, Waltham, MA, USA) with a 96 well block and the following two-step cycler profile: 95 °C for 15 min, 40× (94°C for 15 s, 55°C for 30 s, 70°C for 30 s). Analysis of relative gene expression levels was performed using the formula 2 − ΔCT with ΔCT = CT (target gene) − CT (control). Individual PCR product was analyzed for DNA sequence to verify the purity of the product.

### 2.6. Partial Mouse (A) MALAT1 DNA Fragment (B) MGP DNA Fragment (Contains mir-143-3p Binding Site) Construction

(A)A 500 bp mouse MALAT1 DNA fragment (Malat1-201 ENSMUST00000172812.4 _3385~3884 bp; Chromosome 19: 5,845,717-5,852,704; http://www.ensembl.org/index.html, (accessed on 18 October 2025)) was generated by artificial synthesis and clone was digested with SacI and Xbal restriction enzymes and ligated into pmirNanoGLO plasmid vector (Promega, Madison, WI, USA). The cloned mouse MALAT1 DNA fragment contains miR-143-3p potential binding site (from 3614 bp to 3634 bp). For the mutant, the conserved sites, GAAGATCAAATGTTCATCTCA, were mutated into ACAGATACAATGGGACGAGAC and were constructed by the same method described above. All the cloned plasmids were confirmed by DNA sequencing (Seeing Bioscience Co., Ltd., Taipei, Taiwan).(B)A 209 bp mouse MGP DNA fragment (Mgp-201 ENSMUST00000032342.3 _1~209 bp; Chromosome 6: 136,849,433-136,852,821; http://www.ensembl.org/index.html, (accessed on 18 October 2025)) was generated by artificial synthesis and clone was digested with SacI and Xbal restriction enzymes and ligated into pmirNanoGLO plasmid vector (Promega). The cloned mouse MGP DNA fragment contains miR-143-3p potential binding site (from 26 bp to 46 bp).For the mutant, the conserved sites, GGAGTTTCGTTTTATATCTCC, were mutated into TGAGGTTCTGTTGATCGAGAC and were constructed by the same method described above. All the cloned plasmids were confirmed by DNA sequencing (Seeing Bioscience Co., Ltd.).

### 2.7. Luciferase Activity Assay

The test plasmid (2 μg) was transfected by ViaFect™ Transfection Reagent (Promega) in accordance with the manufacturer’s protocol.The detailed procedure of luciferase activity assay has been described previously [20]. Following treatment, cell extracts were prepared with the Nano-Glo dual-luciferase reporter assay system (Promega) and using a luminometer (Glomax Multi Detection System, Promega) to measure the luciferase activity.

### 2.8. Western Blot Analysis

Cells were collected by scraping and then were centrifuged (300× *g*) for 10 min at 4°C. Thereafter, the pellet was re-suspended and homogenized in a Lysis Buffer (Promega), centrifuging at 10,600× *g* for 20 min. Bio-Rad Protein Assay was used to measure protein content. The detailed procedure of Western blot analysis has been described previously [20]. All Western blots were quantified using densitometry.

### 2.9. Calcium Colorimetric Assay

To quantify the calcium ion concentration in cultured VSMCs or carotid artery, calcium colorimetric assay kit (Sigma-Aldrich, St. Louis, MO, USA)was used. In this assay, the calcium ion concentration was determined by the chromogenic complex formed between calcium ions and *o*-cresolphthalein, which was measured at 575 nm. The procedure of measuring calcium ion content was performed according to the manufacturer’s instructions. The OD value (detected at 575 nm) and protein concentration of samples measured using a Pierce BCA protein assay kit (Thermo Fisher Scientific, Waltham, MA, USA) were used to determine the cell calcium content.

### 2.10. Balloon Injury of the Carotid Artery in Diabetic Rats and Delivery of Exosomes Containing Macrophage-Derived MALAT1

Male Wistar rats (290–310 g), aged 16–18 weeks, purchased from BioLASC (BioLASC Taiwan Co., Ltd., Taipei, Taiwan) were injected with a single dose of intraperitoneal streptozotocin (STZ, Sigma-Aldrich, St. Louis, MO, USA) at 90 mg/kg to induce diabetes. Diabetes was confirmed by the presence of blood glucose concentration of 19 mmol/L for a minimum of one week. One week after STZ injection, the rats were anesthetized with isoflurane (2%) and subjected to balloon catheter injury in the right carotid artery as previously described [21]. Rats were randomly divided into six groups (1) sham (Non-STZ), (2) sham and treatment with macrophage-derived MALAT1-containing exosomes, (3) STZ alone, (4) STZ and treatment with macrophage-derived MALAT1-containing exosomes (70 ug/150 uL), (5) STZ and treatment with MALAT1 siRNA at 500 pmol/L, (6) STZ and treatment with scramble siRNAat 500 pmol/L. Briefly, a 2F Forgarty balloon catheter (Biosensors International, Inc., Newport Beach, CA, USA) was inserted through the right external carotid artery, inflated, and passed three times along the length of the isolated segment (1.5–2 cm in length); after the procedure, the catheter was removed.Thereafter, MALAT1-containing exosomes or MALAT1 siRNA was injected to the segment, and electric pulses were administered with five pulses and five opposite polarity pulses at 250 V/cm, 50 ms duration, 75 ms interval using Parallel fixed platinum electrode (CUY610P2–1, 1 mm tip, 2 mm gap) by CUY21-EDIT Square Wave Electroporator (Nepa Gene, Chiba, Japan). The injected MALAT1-containing exosomes or MALAT1 siRNA was incubated for 10 min. After incubation, unbound MALAT1-containing exosomes or MALAT1 siRNA were aspirated. The carotid artery was then tied off, and the neck wound was closed. At the end of experiment, the rats were sacrificed under anesthesia with an overdose of isoflurane, and the carotid artery was quickly removed and stored in liquid nitrogen. Immunofluorescence staining was performed as described previously [21].The intimal, medial, and adventitial cross-sectional areas were measured using imaging software (Nikon NIS-Elements, version 3.0, Nikon Instruments, Inc., Tokyo, Japan). All animal experiments were approved by the Institutional Animal Care and Use Committee of Shin Kong Wu Ho-Su Memorial Hospital (approval number:SKHACC-01-00-02-07) and followedARRIVE guidelines.

### 2.11. Statistical Analysis

The data are expressed as mean ± SD. Analysis of variance (ANOVA) (InStat 3.1, GraphPad Software Inc., San Diego, CA, USA) was used to determine statistical significance. We used Tukey–Kramer comparison test for pairwise comparisons between multiple groups after ANOVA. Statistical significance was set at *p* < 0.05.

## 3. Results

### 3.1. High Glucose Induced Exosomal MALAT1 Expression in Cultured Macrophages

To evaluate the impact of high glucose on exosomal MALAT1 expression in cultured macrophages, we tested a range of glucose concentrations from 12.5 mM to 100 mM. As shown in Figure 1A, all tested concentrations significantly increased exosomal MALAT1 expression relative to the control, with 25 mM glucose producing the maximal effect. Consequently, 25 mM glucose was selected for subsequent experiments. To exclude the osmotic effect of high glucose on MALAT1 expression, three concentrations of mannitol (25, 100, and 300 mM) were added. Mannitol did not have an effect on MALAT1 expression. To assess the time-dependent effects of high glucose, macrophages were treated with 25 mM glucose for durations ranging from 1 to 6 h. Figure 1B illustrates that exosomal MALAT1 expression was significantly elevated from 1 to 4 h relative to the control, with the highest expression observed at 1 h. In contrast, high glucose treatment did not significantly alter exosomal MALAT1 expression in VSMCs over the same time period.

### 3.2. Macrophage-Derived Exosomes Decreased miR-143-3p and Increased MGP Expression in Cultured Vascular Smooth Muscle Cells

To investigate the effect of macrophage-derived exosomes on MGP expression in cultured VSMCs, varying concentrations of exosomes were added to the culture medium for 3 h. As shown in Figure 2A, macrophage-derived exosomes significantly increased MGP mRNA expression at concentrations ranging from 50 µg to 150 µg relative to the control, with 50 µg producing the maximal effect. Control exosomes at the same concentration did not enhance MGP expression. Therefore, 50 µg of exosomes derived from macrophages treated with 25 mM glucose were used for subsequent experiments. Stimulation with 50 µg of macrophage-derived exosomes for durations ranging from 1 to 6 h significantly enhanced cytoplasmic MALAT1 mRNA levels in cultured VSMCs. However, cytoplasmic miR-143-3p mRNA levels were significantly decreased from 3 to 4 h after treatment. Macrophage-derived exosomes also significantly increased cytoplasmic MGP mRNA expression from 3 to 6 h, exhibiting a pattern similar to that of MALAT1 (Figure 2B).To further investigate the role of MALAT1, we used MALAT1 siRNA to silence its expression. MALAT1 siRNA significantly decreased cytoplasmic MGP mRNA expression induced by macrophage-derived exosomes (Figure 3), whereas scrambled MALAT1 siRNA had no effect. The overexpression of wild-type miR-143-3p significantly inhibited MGP mRNA expression induced by macrophage-derived exosomes, while the overexpression of mutant miR-143-3p or its antagomir did not affect MGP mRNA expression. Macrophage-derived exosomes significantly increased MGP expression from 1 to 3 h compared to the control, with maximal induction observed at 3 h (Figure 4A,B). MALAT1 siRNA significantly decreased cytoplasmic MGP expression induced by macrophage-derived exosomes (Figure 4C,D), whereas scrambled MALAT1 siRNA had no effect. Theoverexpression of wild-type miR-143-3p significantly inhibited MGP expression induced by macrophage-derived exosomes, while the overexpression of mutant miR-143-3p or its antagomir did not affect MGP expression. The purity of macrophage-derived exosomes was confirmed using exosomal biomarkers CD9 and CD81 (Cell Signaling Technology, Danvers, MA, USA) via Western blot analysis. 

### 3.3. MiR-143-3p Decreased MALAT1 and MGP Luciferase Activity in Cultured VSMC Treated with Macrophage-Derived Exosomes

The sequence of the MALAT1 3’UTR target site for miR-143-3p binding (nucleotides 3614–3634) is shown in Figure 5A. The overexpression of miR-143-3p significantly reduced MALAT1 luciferase activity in cultured VSMCs treated with macrophage-derived exosomes when miR-143-3p was bound to the normal MALAT1 3’UTR (Figure 5B). In contrast, the overexpression of mutant miR-143-3p did not significantly alter MALAT1 luciferase activity, indicating that miR-143-3p is a target of MALAT1. We identified a binding site for miR-143-3p in the MGP 3’UTR (nucleotides 26–46), as shown in Figure 5C. The overexpression of miR-143-3p significantly decreased MGP luciferase activity in cultured VSMCs treated with macrophage-derived exosomes when miR-143-3p was bound to the normal MGP 3’UTR (Figure 5D). The overexpression of mutant miR-143-3p did not significantly alter MGP luciferase activity, indicating that MGP is a target gene of miR-143-3p.

### 3.4. MiR-143-3p Increased Calcium Content in Cultured VSMC Treated with Macrophage-Derived Exosomes

The calcium content in VSMCs was not statistically different between macrophage-derived exosome treatment and the control group (Appendix A). However, silencing MALAT1 using MALAT1 siRNA and the overexpression of wild-type miR-143-3p significantly increased calcium content in VSMCs following treatment with macrophage-derived exosomes. In contrast, scrambled siRNA and the overexpression of mutant miR-143-3p or its antagomir did not affect calcium content in VSMCs after treatment with macrophage-derived exosomes.

### 3.5. Carotid Artery Balloon Injury Enhances MALAT1 Expression to Inhibit miR204-5p Expression in Diabetic Rats

MALAT1 mRNA expression was significantly elevated from day 3 to day 28, while miR-143-3p expression was significantly reduced from day 5 to day 28 following balloon injury of the carotid artery in diabetic rats treated with macrophage-derived exosomes, compared to the sham group without treatment (Figure 6). The expression pattern of MGP mRNA in the carotid artery after balloon injury mirrored that of MALAT1 mRNA. MiR-143-3p expression was significantly reduced in diabetic rats treated with or without macrophage-derived exosomes at 14 days post-balloon injury. Silencing MALAT1 by MALAT1 siRNA significantly restored miR-143-3p expression (Figure 7A), whereas scrambled MALAT1 siRNA did not significantly influence miR-143-3p expression as compared to diabetic rats with balloon injury. MGP mRNA and protein expression were significantly increased in diabetic rats treated with or without macrophage-derived exosomes at 14 days post-balloon injury. MALAT1 siRNA and overexpression of wild-type miR-143-3p significantly inhibited MGP mRNA and protein expression compared to diabetic rats treated with macrophage-derived exosomes at 14 days post-balloon injury (Figure 7B–D). In contrast, scrambled siRNA or overexpression of mutant miR-143-3p did not significantly affect MGP mRNA and protein expression. Calcium content in the carotid artery was significantly increased in diabetic rats at 14 days post-balloon injury (Appendix A). Treatment with macrophage-derived exosomes for 14 days post-balloon injury significantly reduced calcium content. However, both MALAT1 siRNA and wild-type miR-143-3p overexpression significantly increased calcium content in the carotid artery following treatment with macrophage-derived exosomes. Both scrambled siRNA and mutant miR-143-3p overexpression did not affect calcium content. MGP fluorescence signals were increased in the carotid artery of diabetic rats and further enhanced upon treatment with macrophage-derived exosomes at 14 days after carotid artery balloon injury, (Appendix A). These signals were decreased by MALAT1 siRNA at 14 days post-balloon injury as compared to the group treated with macrophage-derived exosomes. Alizarin red staining also revealed increased calcium signaling in the carotid artery after balloon injury in diabetic rats treated with macrophage-derived exosomes for 14 days. MALAT1 siRNA further enhanced calcium content in the carotid artery (Appendix A).

## 4. Discussion

In this study, we demonstrate a novel mechanistic insight into how hyperglycemia influences VC via macrophage-derived exosomal signaling. Specifically, we found that the stimulation of macrophages with high glucose (25 mM) significantly upregulated the expression of the long non-coding RNA MALAT1 within exosomes. These exosomes, when delivered to VSMCs, induced a cascade of molecular responses, including the downregulation of miR-143-3p and the upregulation of MGP, a known inhibitor of vascular mineralization. Increased VC following MALAT1 siRNA treatment and miR-143-3p overexpression in cultured VSMCs was confirmed by calcium content analysis using a calcium colorimetric assay. The effect of MALAT1 upregulation was glucose-dependent and not osmotic in effect because mannitol did not have an effect on MALAT1 expression.

Our in vitro experiments provide strong support for a competitive endogenous RNA regulatory model in which exosomal MALAT1 functions as a molecular sponge for miR-143-3p. This interaction prevents miR-143-3p from binding to the 3’ untranslated region (3’UTR) of MGP mRNA, thereby enhancing MGP expression at both the transcript and protein levels. Temporal analysis revealed that exosomal MALAT1 rapidly entered VSMCs and exerted suppressive effects on miR-143-3p within 3 to 4 h, followed by a significant increase in MGP expression from 3 to 6 h. These molecular events were validated using siRNA-mediated MALAT1 knockdown, miR-143-3p overexpression, and luciferase reporter assays, all of which confirmed the specificity of the MALAT1–miR-143-3p–MGP regulatory axis.

Our study results also underscore the role of macrophages not merely as passive inflammatory players [22], but as active mediators of vascular remodeling through their exosomal cargo. Under hyperglycemic stress, macrophages may adopt a distinct phenotype capable of modifying the local vascular microenvironment by secreting bioactive vesicles. The modulation of MGP, a potent anti-calcific protein, via this axis suggests that macrophage-derived exosomes can reprogram VSMC behavior toward a less calcific phenotype—offering a potential molecular checkpoint that could be harnessed therapeutically.

Our in vivo balloon injury model of the carotid artery also demonstrated that macrophage-derived MALAT1 enhanced plaque formation in the carotid artery of diabetic rats. In this animal study, we showed that miR-143-3p expression was reduced and MGP expression was increased following treatment with macrophage-derived exosomes in diabetic rats subjected to carotid artery balloon injury. Silencing MALAT1 using MALAT1 siRNA reversed miR-143-3p expression and inhibited MGP expression in the carotid artery after balloon injury. The increase in VC induced by MALAT1 siRNA in the carotid artery after balloon injury in diabetic rats was confirmed by Alizarin red staining.

MALAT1 has been identified as a regulator of VC and may function as a molecular sponge within the RNA triplet network comprising lncRNA–miRNA–mRNA [13,14]. The MALAT1–miR-143 axis has been implicated in coronary artery in-stent restenosis and may represent a potential pathway for modulating VSMC proliferation [23]. The role of MALAT1 in VC under different stress conditions or across various cell types remains a topic of debate. Gong et al. reported that MALAT1 promoted calcification in human VSMCs under high phosphate conditions [13]. In contrast, our findings suggest that macrophage-derived MALAT1 under hyperglycemic conditions may alleviate VC in cultured adult mouse VSMCs. Further analysis of MALAT1’s effects across different cell types and disease models may help reconcile these seemingly contradictory reports regarding MALAT1-mediated promotion and attenuation of VC.

## 5. Conclusions

This study identifies a previously unrecognized exosomal lncRNA-mediated pathway involving MALAT1, miR-143-3p, and MGP that regulates the response of vascular smooth muscle cells (VSMCs) to hyperglycemia. Our findings offer novel insights into the intercellular communication between macrophages and VSMCs, opening new avenues for RNA-based diagnostics and therapeutics in diabetes-associated vascular disease.

## Figures and Tables

**Figure 1 cells-14-01995-f001:**
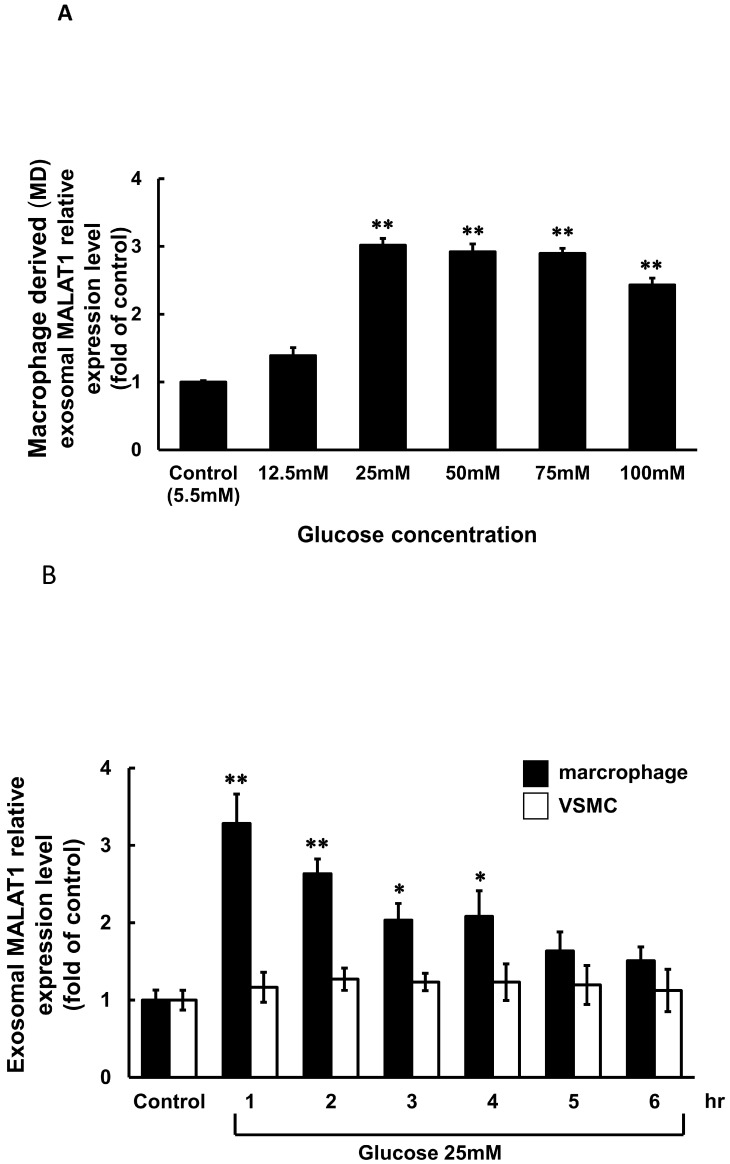
Effect of high glucose on exosomal MALAT1 expression in cultured macrophages and vascular smooth muscle cells (VSMCs). (**A**) Treatment with different glucose concentrations for 1 h. (**B**) Treatment with a glucose concentration of 25 mM for different periods of time. * *p* < 0.01 vs. control. ** *p* < 0.001 vs. control. *N* = 4 per group.

**Figure 2 cells-14-01995-f002:**
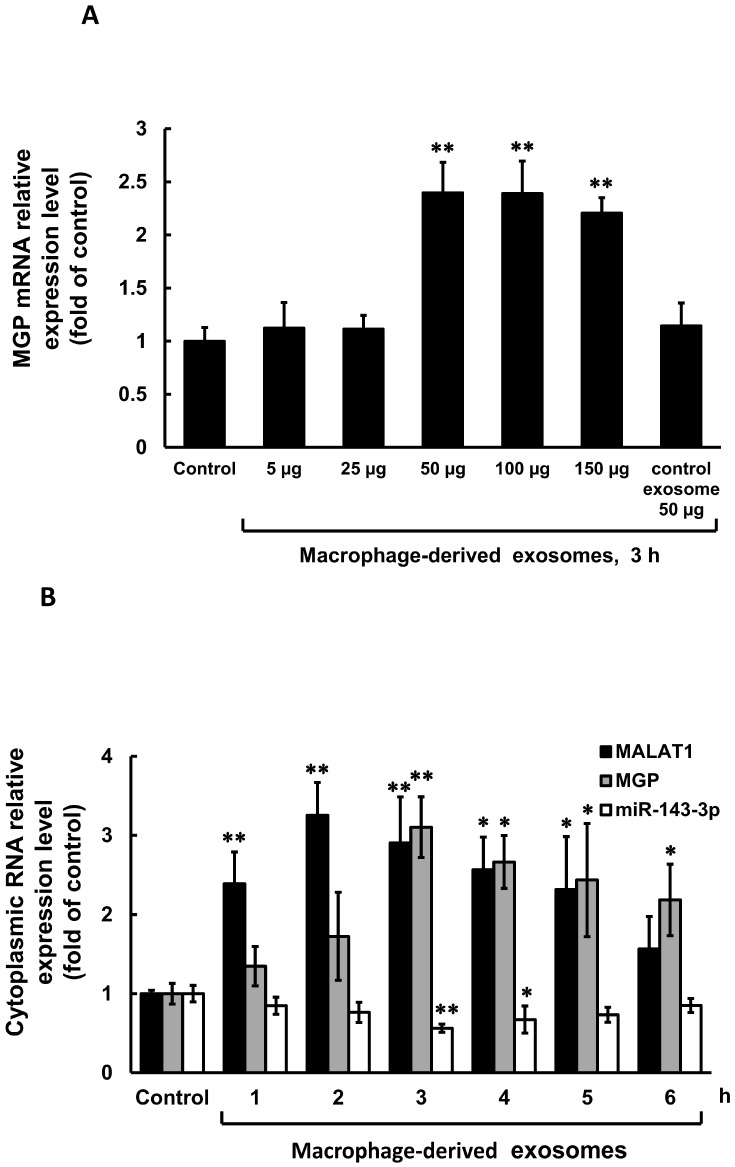
Effect of macrophage-derived exosomes on MALAT1, miR-143-3p, and MGP cytosolic mRNA expression in cultured VSMCs. (**A**) Quantitative real-time PCR of cytosolic MGP mRNA with different doses of macrophage-derived exosomes. (**B**) Quantitative real-time PCR of cytoplasmic MALAT1, miR-143-3p, and MGP mRNA levels. A measurement of 50 μg of macrophage-derived exosomes extracted from macrophages under 25 mM glucose stimulation for 3 h was used for the treatment group. * *p* < 0.01 vs. control. ** *p* < 0.001 vs. control. *N* = 4 per group.

**Figure 3 cells-14-01995-f003:**
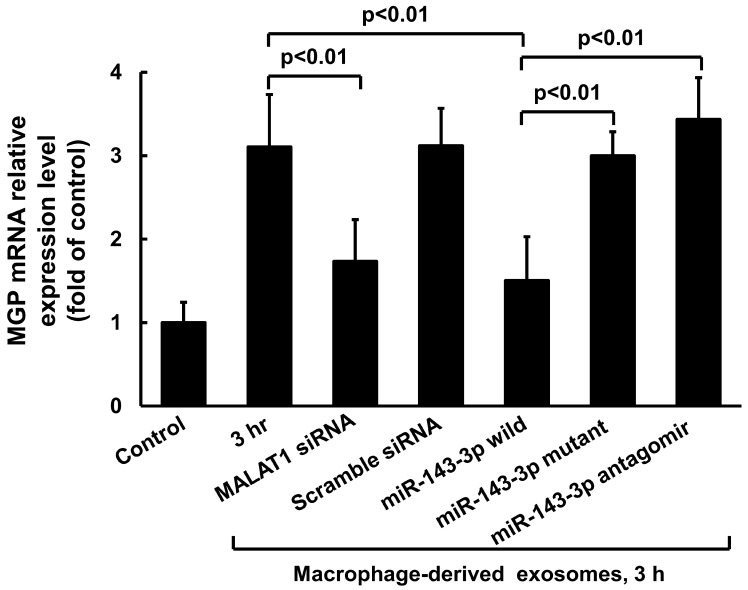
Effect of macrophage-derived exosomes on MGP cytosolic mRNA expression with different treatment in cultured VSMCs. Quantitative real-time PCR of cytosolic MGP mRNA with different treatment. A measurement of 50 μg of macrophage-derived exosomes extracted from macrophages under 25 mM glucose stimulation for 3 h was used for the treatment group. *N* = 4 per group.

**Figure 4 cells-14-01995-f004:**
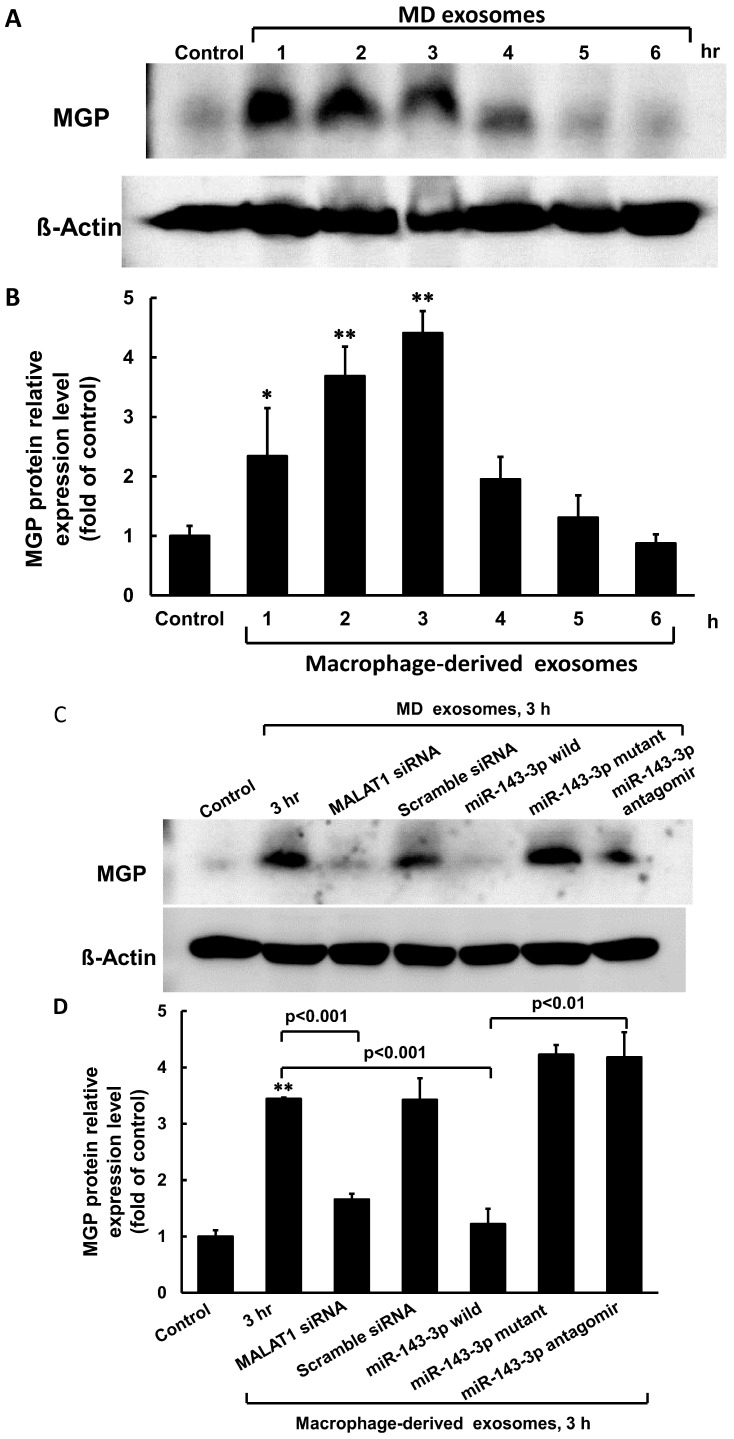
Effect of macrophage-derived exosomes on MGP expression with different treatment in cultured VSMCs. (**A**) Representative Western blots for MGP and ß-actin protein levels in VSMCs under 25 mM glucose treatment for different periods of time. (**B**) Quantitative analysis of MGP levels. The values for stimulated VSMCs have been normalized to the control cell values. (**C**) Representative Western blots for MGP and ß-actin protein levels in VSMCs under 25 mM glucose treatment with different treatment. (**D**) Quantitative analysis of MGP levels. The values for stimulated VSMC have been normalized to the control cell values. * *p* < 0.01 vs. control. ** *p* < 0.001 vs. control. *N* = 3 per group.

**Figure 5 cells-14-01995-f005:**
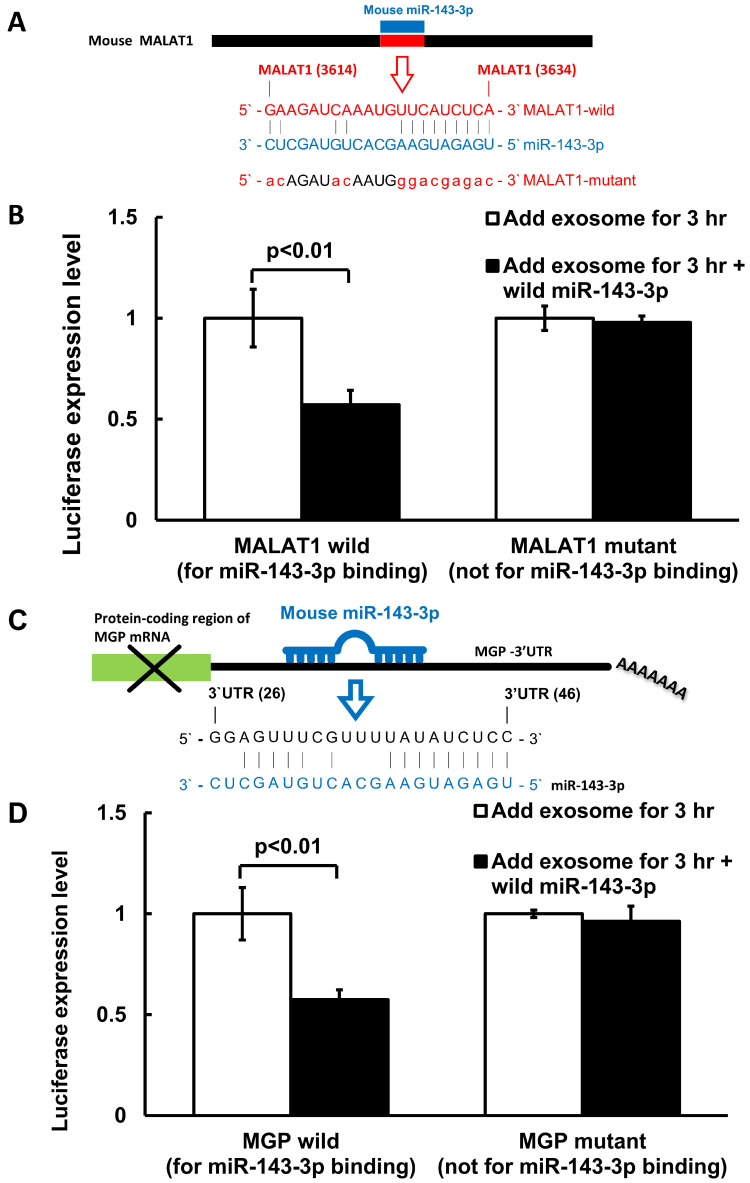
Macrophage-derived exosomes affect miR-143-3p on MALAT1 and MGP luciferase activity in cultured VSMC. (**A**)Nucleotides 3614 to 3634 bp sequence of the mouse MALAT1 3’UTR target site for miR-143-3p binding. (**B**) MALAT1 3’UTR luciferase activity under macrophage-derived exosomes treatment with wild-type or mutant MALAT1. (**C**) Nucleotides 26 to 46 bp sequence of the mouse MGP 3’UTR target site for miR-143-3p binding, which is located on the MGP3’UTR. (**D**) MGP3’UTR luciferase activity under macrophage-derived exosomes treatment with wild-type or mutant MGP. *N* = 4 per group.

**Figure 6 cells-14-01995-f006:**
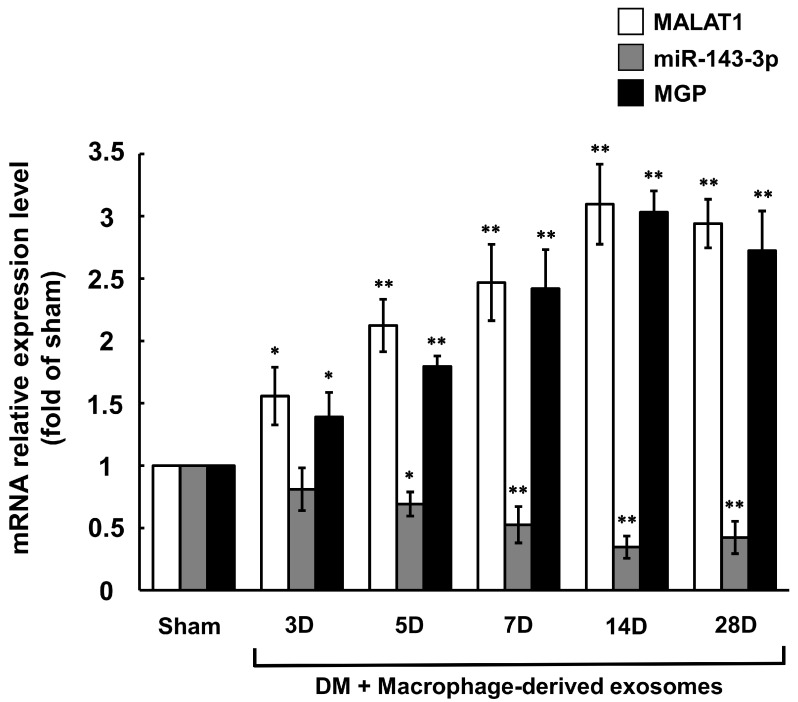
Effect of macrophage-derived exosomes on MALAT1, miR-143-3p, and MGP mRNA expression after right external carotid artery balloon injury in diabetic rats. Quantitative real-time PCR analysis of MALAT1, miR-143-3p, and MGP mRNA levels in carotid artery with or without carotid artery balloon injury in diabetic rats after different periods of time. * *p* < 0.01 vs. control. ** *p* < 0.001 vs. control. *N* = 5 per group.

**Figure 7 cells-14-01995-f007:**
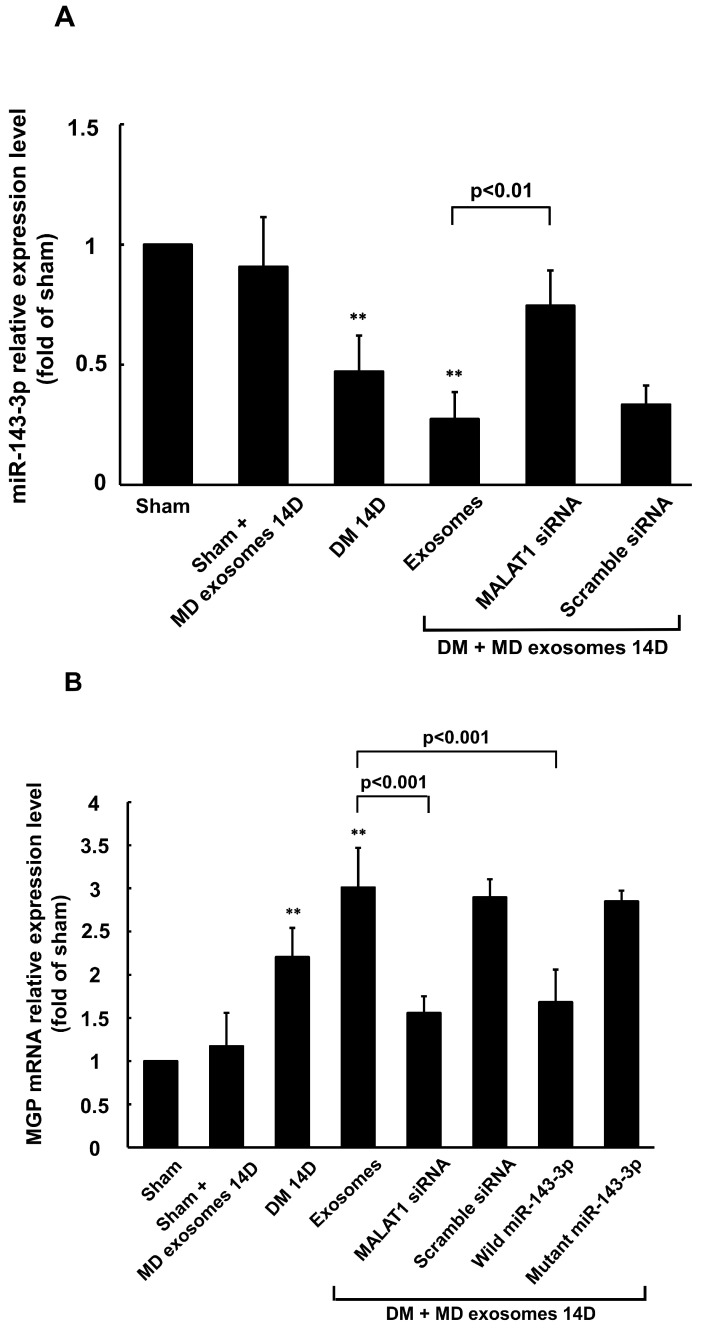
Macrophage-derived exosomal MALAT1mediates the reduction in miR-143-3p and the induction of MGP expression after carotid artery balloon injury in diabetic rats. (**A**)Quantitative real-time PCR analysis of miR-143-3p mRNA levels in arterial tissue at 14 days after carotid artery balloon injury in diabetic rats. The macrophage-derived exosomes were extracted from macrophages under 25 mM glucose treatment for 3 h. Scramble siRNA was the control siRNA. (**B**)Quantitative real-time PCR of MGP mRNA levels in arterial tissue at 14 days after carotid artery balloon injury in diabetic rats. (**C**) Representative Western blots for MGP and α-tubulin protein levels in carotid artery at 14 days after right external carotid artery balloon injury in diabetic rats. (**D**) Quantitative analysis of MGP levels. The values for MGP have been normalized to the sham group. The macrophage-derived exosomes were extracted from macrophages under 25 mM glucose stimulation for 3 h. ** *p* < 0.001 vs. sham. *N* = 5 per group.

## Data Availability

The data presented in this study are available on request from the corresponding author.

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
