# Peer review of "Macrophage-Derived Exosomal MALAT1 Induced by Hyperglycemia Regulates Vascular Calcification Through miR-143-3p/MGP Axis in Cultured Vascular Smooth Muscle Cells and Diabetic Rat Carotid Artery"

_cells, 2025, doi:10.3390/cells14241995_

Round 1
Reviewer 1 Report
Comments and Suggestions for Authors
Dear Editor,
I have read with extreme interest the paper by Dr. Kou-Gi et. al etitled “Macrophage-derived exosomal MALAT1 induced by hypergly-2 cemia regulates vascular calcification through miR-143-3p/MGP 3 axis in cultured vascular smooth muscle cells and diabetic rat 4 carotid artery“.
In general, accumulating studies indicate so far that long non-coding RNAs (lncRNAs) play important roles in the regulation of diverse biological processes involved in homeostatic control of the vessel wall in health and disease. However, our knowledge of the mechanisms by which lncRNAs control gene expression and cell signaling pathways is still nascent. Furthermore, only a handful of lncRNAs has been functionally evaluated in response to pathophysiological stimuli or in vascular disease states. For example, lncRNAs may regulate endothelial and vascular smooth muscle cells dysfunction by modulating endothelial cell proliferation (e.g. MALAT1, H19) or angiogenesis (e.g. MEG3, MANTIS). LncRNAs have also been implicated in modulating vascular smooth muscle cell (VSMC) phenotypes or vascular remodeling (e.g. ANRIL, SMILR, SENCR, MYOSLID). Finally, emerging studies have implicated lncRNAs in leukocytes activation (e.g. lincRNA-Cox2, linc00305, THRIL), macrophage polarization (e.g. GAS5), and cholesterol metabolism (e.g. LeXis).
The findings of this paper are totally in line with this scenario and shed additional lights on it providing new opportunities for the generation of a new class of RNA-based biomarkers and therapeutic targets.
In particular authors found a “novel mechanistic insight into how hyperglycemia influences vascular calcification (VC) via macrophage-derived exosomal signaling. Specifically, they found that stimulation of macrophages with high glucose (25 mM) significantly upregulated the expression of the long non-coding RNA MALAT1 within exosomes. These exosomes, when delivered to VSMCs, induced a cascade of molecular responses, including the downregulation of miR-143-3p and the upregulation of MGP, a known inhibitor of vascular mineralization. Increased VC following MALAT1 siRNA treatment and miR-143-3p overexpression in cultured VSMCs was confirmed by calcium content analysis using a calcium colorimetric assay”.
The paper was elegantly written and data are presented in a concise yet exhaustive manner making the reading experience very pleasant.
Author Response
Thank you for the kind and positive review to our manuscript.
Reviewer 2 Report
Comments and Suggestions for Authors
The authors investigated mechanisms of vascular calcification (VC) in conditions of hyperglycemia, diabetes and vascular injury in vitro and in vivo. They found that cultured macrophages in high glucose media produce exosomes containing increased levels of MALAT1, which induces production of MGP in vascular smooth muscle cells (VSMC) in vitro. MGP inhibits production of miR-143-3p, which promotes VC. They verified this process in diabetic rats after balloon injury of the carotid artery. Thus, hyperglycemia appears to promote a process in macrophages that inhibits VC. Yet, VC increased in diabetic rats after balloon injury, indicating that the inhibitory process is ineffective in vivo. These results seem to recapitulate the controversy surrounding the role of VC in atherosclerosis, with much evidence suggesting that calcification stabilizes atheroma and is not as pathologic as is widely believed. It also mirrors the questionability of exosomal factors in pathological processes. Perhaps exosomal MALAT1 is irrelevant to VC in vivo. The authors need to present findings that contribute to elucidating and resolving the contradictions of this study. Specific comments: The minimum glucose concentration tested in the Methods is 5 mM, but the concentration range in the Results is 25-100 mM and Figure 1A shows a control concentration of 5.5 mM. These discrepancies should be resolved. MGP should be defined in the abstract.
Author Response
- We have corrected the discrepancies. The control glucose concentration is 5.5 mM. Different glucose concentrations from 12. 5 mM to 100 mM were used.
- MGP has been defined in the abstract as “Matrix Gla protein (MGP, an inhibitor of VC).
Reviewer 3 Report
Comments and Suggestions for Authors
The authors examined the effects of exosomal MALAT1 derived from macrophages induced by hyperglycaemia on vascular calcification (VC). Endothelial dysfunction and inflammation further modulate exosome composition and release, enhancing calcific signalling. Crosstalk between ECs, VSMCs, and adventitial fibroblasts via exosomes contributes to the propagation of calcification throughout the vascular wall. Moreover, exosomes derived from macrophages in atherosclerotic plaques can exacerbate calcification by promoting VSMC osteogenic differentiation. The authors are investigating a timely area of research that could yield novel regulatory mechanisms in hyperglycaemia-induced vascular damage.
The authors have taken a cultured cell approach, investigating the effects of macrophage-derived exosomes from culture conditions. These exosomes were assessed against cultured vascular smooth cells obtained from PromoCell.
This a very interesting proposal that has the potential to yield some interesting mechanistic insight into a poorly understood mechanisms in vascular disease.
Comments:
- This reviewer fully appreciates the need to test across a range of glucose for an effect. 100 mM is a vast concentration to try, and very possibly in a severely pathological concentration that would be rarely seen in man. Could the authors comment on the whether they consider the effect to be glucose-dependent, or osmotic in effect? There is no obvious discussion of an osmotic control, such as balancing with mannitol or other non-metabolised sugar.
- The authors should also discuss the chosen “hyperglycaemic” concentration of 25 mM. This is also a rather extreme level of hyperglycaemia in a patient population. I would expect that this concentration has been chosen as many commercial media recipes have this as a standard level. Stress hyperglycaemia is typically considered to be in the range of 11 – ~17 mM. It is helpful for this study that the authors include a concentration of 12/5 mM, as there is a change in the exosome release and quantity, helping to provide some more physiologically/pathophysiological relevance (for a larger quantity of patients). A time course in a lower concentration would also be worth investigating. Furthermore, what are the control experiments here? Are they an equivalent increase in osmolarity, with mannitol/alternative osmotic balance, or just a culture with no change in bathing glucose? I am not convinced that the lack of media supplementation be an appropriate control in this case, given the osmotic difference in 25 mM vs 5 mM glucose solution.
- In figures 6 and 7, where the data is normalised to the levels in Sham, is this normalised to the mean data? It isn’t clear, given that there is an error bar expressed, what the others have been normalised to.
- The levels have been normalised alpha-tubulin. Are the authors convinced that that there are no alterations to the levels of their housekeeping gene over this time course? Given that this model of diabetes is requiring animals to have a blood glucose greater and 19 mM over a weeklong period, what other phenotypic changes are seen in the vasculature? Is the blood glucose regulation in sham surgery rats equivalent to those that have been through the balloon injury protocol?
- The methods state that the animals are culled by terminal anaesthetic after the catheter injury. However, the data presented in the figures state that the analysis of mrR-143-3p and other components were taken at 14 days (figure 7), or up to 28 days (figure 6) in macrophage-derived exosome treatment. I suspect that there is a step missing in the methods section that describes the delivery of the exosomes and the amounts added, “One week after STZ injection, the rats were anesthetized with isoflurane (2%) and subjected to balloon catheter injury in the right carotid artery as previously described [21]. At the end of experiment the rats were sacrificed under anesthesia with an overdose of isoflurane, and the carotid artery was quickly removed…”. Some clarification is needed here.
Minor Comment:
Figure 1B – mislabelled as “Marcophage” not Macrophage.
Comments on the Quality of English LanguageThere are some small examples of poor English throughout the manuscript, but nothing that I found difficult to follow.
Author Response
- To exclude the osmotic effect of high glucose on MALAT1 expression, three concentrations of mannitol (25, 100, and 300 mM) were added. Mannitol did not have the effect on MALAT1 expression. The above statement has been added in the Result section 3.1 on page 13. The following statement “The effect of MALAT1 upregulation was glucose-dependent and not osmotic in effect because mannitol did not have the effect on MALAT1 expression.” has been added in the Discussion section on page 19.
- Glucose concentration at 20-30 mM has been commonly used for in vitro study such as smooth muscle cell and endothelial cell. As reviewer claimed that stress hyperglycaemia is typically considered to be in the range of 11 – ~17 mM. The glucose concentration used in the in vitro study is usually much higher than that in the patient status. There is no osmotic effect on MALAT1 expression.
- We have corrected the error in Figure 6 and Figure 7. The bar of the sham group has been removed.
- There are no alterations to the levels of alpha-tubulin housekeeping gene over the time course. The phenotypic change of the vasculature of diabetic rat is intimal area increase and calcium content increase. The blood glucose regulation in sham surgery rats is equivalent to those that have been through the balloon injury protocol.
- We have clarified the animal study as “Rats were randomly divided into six groups (1) sham (Non-STZ), (2) STZ alone, (3) STZ and treatment with macrophage-derived MALAT1-contained exosomes (70ug/150uL), (5) STZ and treatment with MALAT1 siRNA at 500pmol/L, (6) STZ and treatment with scramble siRNA at 500pmol/L. Briefly, a 2F Forgarty balloon catheter (Biosensors International, Inc., Newport Beach, CA, USA) was inserted through the right external carotid artery, inflated and passed three times along the length of the isolated segment (1.5–2 cm in length); then the catheter was removed. MALAT1-contained exosomes or MALAT1 siRNA was injected to the segment, and electric pulses using CUY21-EDIT Square Wave Electroporator (Nepa Gene, Chiba, Japan)) were administered with five pulses and five opposite polarity pulses at 250 V/cm, 50 ms duration, 75 ms interval using Parallel fixed platinum electrode (CUY610P2–1, 1 mm tip, 2 mm gap). The injected MALAT1-containing exosomes or MALAT1 siRNA was incubated for 10 min. After incubation, unbound MALAT1-containing exosomes or MALAT1 siRNA was aspirated. The carotid artery was then tied off, and the wound was closed.” The above statement has been added in the section 2.10 on page 11 and 12.
- In figure 1B, mislabeled “Marcophage” has been corrected as “Macrophage”.
Round 2
Reviewer 2 Report
Comments and Suggestions for Authors
My concerns have been addressed. I have no further comments.
Reviewer 3 Report
Comments and Suggestions for Authors
The authors have address my major concerns. I have no further comments to make on this.